# COOPERATIVE ADVERSARIAL LEARNING VIA CLOSED-LOOP TRANSCRIPTION

## ABSTRACT

Generative models based on the adversarial process are sensitive to net architectures and difficult to train. This paper proposes a generative model that implements cooperative adversarial learning via closed-loop transcription. In the generative model training, the encoder and decoder are trained simultaneously, and not only the adversarial process but also a cooperative process is included. In the adversarial process, the encoder plays as a critic to maximize the distance between the original and transcribed images, in which the distance is measured by rate reduction in the feature space; in the cooperative process, the encoder and the decoder cooperatively minimize the distance to improve the transcription quality. Cooperative adversarial learning possesses the concepts and properties of Auto-Encoding and GAN, and it is unique in that the encoder actively controls the training process as it is trained in both learning processes in two different roles. Experiments demonstrate that without regularization techniques, our generative model is robust to net architectures and easy to train, sample-wise reconstruction performs well in terms of sample features, and disentangled visual attributes are well modeled in independent principal components.

## 1 INTRODUCTION

Minimax game provides an unsupervised learning method, which is widely used in generative models such as generative adversarial nets (GAN) (Goodfellow et al., 2014; Chen et al., 2016; Radford et al., 2015) and the recently-proposed closed-loop transcription framework (CTRL) (Dai et al., 2022). Generative modeling based on minimax two-player game faces some problems, like the instability in training processes, the difficulty in maintaining the balance between the discriminator and the generator (as in GAN) or between the encoder and the decoder (as in CTRL), and the sensitiveness to net architectures (He et al., 2016a;b).

Maintaining balance and stability in the adversarial process attracts a lot of attention. The mainstream is to provide a constrained discriminator (Kurach et al., 2019). Some regularization techniques are provided, such as weight normalization (Salimans & Kingma, 2016), weight clip (Arjovsky et al., 2017), gradient penalty (Gulrajani et al., 2017), spectral normalization (Miyato et al., 2018), and adversarial lipschitz regularization (Terjék, 2019).

Different from the mainstream regularization methods, this paper considers the feasibility of letting the discriminator actively adapt to the rhythm of the generator. The reason why maintaining balance in the generative models via adversarial process is difficult is that the generator and the discriminator tend to merely play against each other. However, balance will break sooner or later once the discriminator learns faster than the generator. In contrast, generative models based on Auto-Encoding like variational Auto-Encoding (VAE) (Kingma & Welling, 2013; Lopez et al., 2018) tend to be stable, not facing instability and collapse problems. The reason is that the encoder and decoder in the Auto-Encoding framework learn and update themselves cooperatively to improve reconstruction quality and reduce data dimensions in the same direction. In one word, models work cooperatively rather than against each other. Inspired by this cooperation idea, this paper attempts to combine cooperative learning and adversarial learning in the generative model.

In this paper, a generative model via cooperative adversarial learning (CoA-CTRL) is proposed. CoA-CTRL employs the closed-loop transcription framework (CTRL) proposed by (Dai et al., 2022; Ma et al., 2022) and naturally combines the learning strategies of the adversarial process and cooperative process. Firstly, like the discriminator in GAN, the encoder in CoA-CTRL plays as a critic to

maximize the feature distance between the real data and the transcribed data. Secondly, consistent with Auto-Encoding, the encoder and decoder cooperatively minimize the difference between the real data and transcribed data. The confrontation and cooperation between the two take turns and intersect, which will actively keep the system in balance.

## 2 RELATED WORK

**Auto-Encoding and its variants.** Auto-Encoding is a typical neural network for representation learning and data dimension reduction (Kramer, 1991; Hinton & Zemel, 1993; Hinton & Salakhutdinov, 2006). Auto-Encoding aims to learn the encoder $E_\theta$ and the decoder $D_\eta$ simultaneously, and this process can be demonstrated by equation (1). Generally, Auto-Encoding tends to learn from L2 pixel-wise distance.

$$\min_{\theta,\eta} \mathcal{L}(\theta, \eta) = \frac{1}{N} \sum_{i=1}^{N} ||x_i - E_\theta(D_\eta(x_i))||_2^2 \tag{1}$$

**Generative adversarial nets (GAN).** Generative adversarial nets (GAN) provides a generative model based on the adversarial process (Goodfellow et al., 2014; Chen et al., 2016). GAN includes a discriminator and a generator. The discriminator evaluates the performance of the generated images, and the generator tends to fool the discriminator. The two networks are trained based on the two-player minimax game by the value function $V(G(\eta), D(\theta))$ as equation (2) displays, where $G(\eta)$ and $D(\theta)$ donate to the generator and discriminator respectively.

$$\min_{\eta} \max_{\theta} V(\eta, \theta) = \mathbb{E}_{x \sim p(x)}[\log D(x)] + \mathbb{E}_{z \sim p(z)}[\log(1 - D(G(z)))] \tag{2}$$

**MCR$^2$ and CTRL**. Recently, Chan et al. (2022) and Yu et al. (2020) proposed a new learning objective, the so-called principle of maximal coding rate reduction (MCR$^2$), which is to learn the low-dimension intrinsic structures from high dimension data and obtain discriminative representation between classes. Encoder $f(x, \theta)$ maps high dimension data $X$ to the low dimension features $Z$. As is shown in equation (3), MCR$^2$ provides a method called coding rate ($R(Z, \epsilon)$) to measure the compactness of learned feature $Z$ integrally subject to the distortion $\epsilon$. The rate reduction ($\Delta R$) measures the distance in the feature space. In a special case of two classes [1], as shown in equation (6), there will be data features $Z$ and $\hat{Z}$. The distance between $Z$ and $\hat{Z}$ can be measured by coding rate reduction ($\Delta R(Z, \hat{Z})$), that is, the difference between the coding rate of ($Z \cup \hat{Z}$) and the average sum of them ($R_c$).

$$R(Z, \epsilon) = \frac{1}{2} \log \det(I + \alpha Z Z^*) \tag{3}$$

$$\underbrace{X \xrightarrow{f(x,\theta)} Z \xrightarrow{g(z,\eta)} \hat{X} \xrightarrow{f(x,\theta)} \hat{Z}}_{h(x,\theta,\eta)=f \circ g \circ f} \tag{4}$$

$$h(x, \theta, \eta) = f(g(f(x, \theta), \eta), \theta) \tag{5}$$

$$\Delta R(Z, \hat{Z}) = R(Z \cup \hat{Z}) - \underbrace{\frac{1}{2}(R(Z) + R(\hat{Z}))}_{R_c} \tag{6}$$

$$\min_{\eta} \max_{\theta} \mathcal{T}(\theta, \eta) = \Delta R(f(X, \theta), h(X, \theta, \eta)) = \Delta R(Z(\theta), \hat{Z}(\theta, \eta)) \tag{7}$$

CTRL (Dai et al., 2022) provides a closed-loop framework based on MCR$^2$, consisting an encoder ($f(x, \theta)$) and a decoder ($g(z, \eta)$). As equation (7) shows, CTRL aims to transcribe data via mini-maxing coding rate reduction, in which $h(x, \theta, \eta)$ captures a closed-loop map as demonstrated by equations (4) and (5). The first segment ($x \rightarrow z \rightarrow \hat{x}$) in (4) resembles Auto-Encoding, and the second segment ($z \rightarrow \hat{x} \rightarrow \hat{z}$) resembles GAN. While GAN generates images from Random Gaussian Distribution noise, in CTRL, as (4) displays, decoder $g(z, \eta)$ maps from feature $Z$ (which is encoded from $X$), and then encoder $f(x, \theta)$ maps $\hat{X}$ to feature $\hat{Z}$. The distance between $X$

---

[1]Where $X \in \mathbb{R}^{D \times n}$ refers to data samples, $Z \in \mathbb{R}^{d \times n}$ refers to features, $\alpha = \frac{d}{n\epsilon^2}$

and $\hat{X}$ is described by the coding rate reduction ($\Delta R$) in equation (6). $R(Z \cup \hat{Z})$ refers to the coding rate of joint space of $Z$ and $\hat{Z}$. $R_c$ describes the average sum coding rate of $Z$ and $\hat{Z}$. The encoder maximizes $\Delta R$, and the decoder minimizes it. In this adversarial learning process, CTRL is consistent with GAN (Goodfellow et al., 2014).

Chan et al. (2022) shows that the gradient of $\Delta R$ will disappear when $\Delta R$ is large enough, which leads to low learning efficiency of encoder $f(x, \theta)$ and decoder $g(z, \eta)$. The $\Delta R$ (the yellow ball) in the left part of Figure 1 demonstrates this situation. This is consistent with Arjovsky & Bottou (2017), which shows that when the distribution of real images and fake images does not intersect in GAN, the gradient of the generator disappears. Therefore, keeping $\Delta R$ at a small value is important. The $\Delta R$ of the right part of Figure 1 is better than the left part for the minimax game.

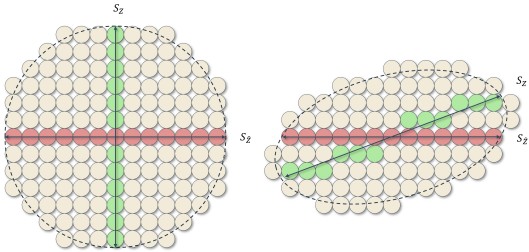

Figure 1: Two different representations compared based on different rate reductions (Yu et al., 2020; Chan et al., 2022). $R(Z \cup \hat{Z})$ is demonstrated by a number of $\epsilon$-balls in the joint space of $Z$ and $\hat{Z}$ (all the balls). $R_c$ is the sum of a number of subspaces of $Z$ (green ball) and $\hat{Z}$ (red ball). $\Delta R$ describes their difference (yellow ball). While MCR$^2$ prefers the left representation for large rate reduction, in the minimax game, the right is better.

## 3 Cooperative adversarial learning

### 3.1 Closed-loop transcription: one encoder, two roles

The way closed-loop transcription combines the structures of Auto-Encoding and GAN is ingenious, as it gives the encoder two different roles. As the left part of Figure 2 and equation (4) demonstrates, in the segment of $x \to z \to \hat{x}$, the encoder takes the responsibility as the encoder in Auto-Encoding; while in the segment of $z \to \hat{x} \to \hat{z}$, the encoder takes the responsibility consistent with the discriminator in GAN. Different from the former works such as VAE-GAN (Larsen et al., 2016) and BiGAN (Donahue et al., 2016; Dumoulin et al., 2016) who add a discriminator to estimate the decoder, closed-loop transcription trains only the encoder and the decoder, and it is the encoder that estimates the performance of the decoder.

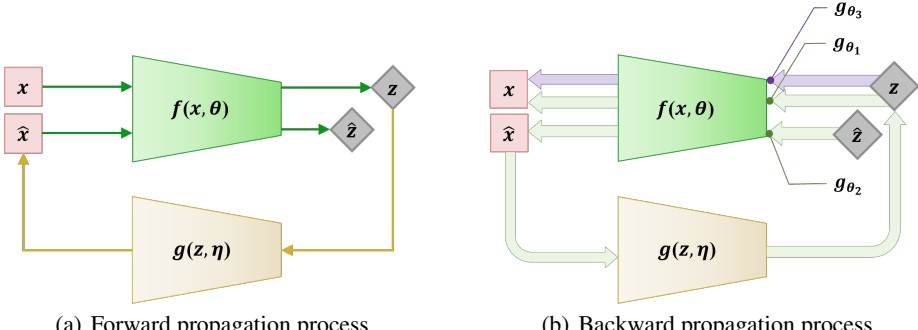

(a) Forward propagation process          (b) Backward propagation process

Figure 2: Forward propagation process (a) and backward propagation process (b) of CTRL. In the backward propagation process, encoder $f(x, \theta)$ was back-propagated three times, and therefore generated three gradient values $g_{\theta 1}$, $g_{\theta 2}$, and $g_{\theta 3}$ respectively.

### 3.2 COOPERATIVE ADVERSARIAL LEARNING: TWO ROLES, LEARN TWICE

**Detailed analysis of the encoder's two roles.** CTRL's two-role setting and closed-loop framework contribute to the complexity of the objective distance function $\Delta R(Z, \hat{Z})$ as well as the training process. As shown in equations (4) and (5), in CTRL's closed-loop map $H(x, \theta, \eta)$, the encoder $f(x, \theta)$ is used twice. We can expand the closed-loop map as shown in equation (8), in which $\hat{Z}(\theta, \eta)$ is expanded to $\hat{Z}(\theta_1, \eta, \theta_2)$. We use $\theta_1$ and $\theta_2$ to refer to the encoder's first-time usage $(X \rightarrow Z)$ and second-time usage $(\hat{X} \rightarrow \hat{Z})$. We can also expand the distance function $\Delta R$ as shown in equation (9), in which $\Delta R(Z, \hat{Z})$ is expanded to $\Delta R(Z(\theta_3), \hat{Z}(\theta_1, \eta, \theta_2))$. $\theta_1$, $\theta_2$, and $\theta_3$ all refer to the encoder $f(x, \theta)$, but we mark it differently as it appears three times and its meanings in the distance function $\Delta R(Z(\theta_3), \hat{Z}(\theta_1, \eta, \theta_2))$ are different. $\theta_2$ and $\theta_3$ represent the encoder when it functions as a discriminator to transcribe data $\hat{X}$ and $X$, and $\theta_1$ represents the encoder when it takes the responsibility of an encoder in the segment of $X \rightarrow Z \rightarrow \hat{Z}$.

$$H(X, \theta, \eta) = f(g(f(X, \theta), \eta), \theta) = \hat{Z}(\theta, \eta) = \hat{Z}(\theta_1, \eta, \theta_2) \tag{8}$$

$$\Delta R(Z, \hat{Z}) = \Delta R(Z(\theta_3), \hat{Z}(\theta_1, \eta, \theta_2)) \tag{9}$$

In the backward propagation process, as shown in the right part of Figure 2, encoder $f(x, \theta)$ will be calculated the gradient three times by the objective function $\Delta R(Z(\theta_3), \hat{Z}(\theta_1, \eta, \theta_2))$. The three gradients are $g_{\theta_1}$, $g_{\theta_2}$, and $g_{\theta_3}$, which are shown in equations (10), (11), and (12).

$$g_{\theta 1} = \nabla_{\theta 1} \Delta R(Z(\theta_3), Z(\theta_1, \eta, \theta_2)) \tag{10}$$

$$g_{\theta 2} = \nabla_{\theta 2} \Delta R(Z(\theta_3), Z(\theta_1, \eta, \theta_2)) \tag{11}$$

$$g_{\theta 3} = \nabla_{\theta 3} \Delta R(Z(\theta_3), Z(\theta_1, \eta, \theta_2)) \tag{12}$$

**Cooperative adversarial learning based on encoder's two roles.** Although CTRL gives the encoder two roles, the original CTRL does not make use of the encoder's dual identity features in terms of its learning strategy (see Algorithm 2 in the Appendix). The original CTRL just follows the simple minimax game in which the encoder functions merely as a discriminator, and its original role of an encoder is ignored. In this paper, we provide cooperative adversarial learning as shown in Algorithm 1 which makes use of the encoder's two roles (encoder and discriminator).

Algorithm 1 demonstrates cooperative adversarial learning. Cooperative adversarial learning comprises two processes:

**(1) Adversarial process**. We only use gradients $g_{\theta 2}$ and $g_{\theta 3}$ (when the encoder functions as a discriminator) in the adversarial process to maximize $\Delta R$ to provide an iteration learning signal. As shown in Algorithm 1, via the adversarial process, encoder $\theta$ updates itself by its role of discriminator via $\theta_2$ (facing input $\hat{X}$) and $\theta_3$ (facing input $X$) to enlarge $\Delta R$. This process is consistent with GAN and the original CTRL, and the equation is demonstrated by equation (13).

**(2) Cooperative process**. We use $g_{\theta 1}$ (when the encoder functions as the encoder in Auto-encoding), $g_{\theta 2}$, $g_{\theta 3}$, and $g_\eta$ together in the cooperative process to compress and transcribe data following the learning signal of $\Delta R$. As shown in Algorithm 1, via the cooperative process, we optimize $\theta_1$, $\theta_2$, $\theta_3$, and $\eta$, which are all elements in the closed-loop transcription, to compress data and transcribe data via the learning signal of $\Delta R$. Equation (14) demonstrates this process.

$$\max_\theta \Delta R(f(X, \theta), H(X, \theta, \eta)) = \Delta R(Z(\theta), \hat{Z}(\theta, \eta)) \tag{13}$$

$$\updownarrow adversarial$$

$$\min_{\theta, \eta} \Delta R(f(X, \theta), H(X, \theta, \eta)) = \Delta R(Z(\theta), \hat{Z}(\theta, \eta)) \tag{14}$$

$$\underbrace{\phantom{\min_{\theta, \eta}}}_{cooperation}$$

---

**Algorithm 1** Cooperative Adversarial Learning

---

**Require:** $\alpha$, learning rate. $ratio$, CoA-ratio. $\epsilon^2$, coding rate parameter. $bs$, batch size.
**Require:** $\theta$, init parameters of the encoder. $\eta$, init parameters of the decoder.
  **while** $\eta$ has not converged **do**
      **Adversarial process to provide iteration learning signal** $\triangle R$
      Sample $X \leftarrow {x^{(i)}}_{i=1}^{bs}$ a batch from the real data
      $Z = f(x, \theta), \hat{X} = (Z, \eta), \hat{Z} = f(\hat{X}, \theta)$
      $g_{\theta 2} \leftarrow \nabla_{\theta 2} \triangle R(Z(\theta_3), \hat{Z}(\theta_1, \eta, \theta_2))$
      $g_{\theta 3} \leftarrow \nabla_{\theta 3} \triangle R(Z(\theta_3), \hat{Z}(\theta_1, \eta, \theta_2))$
      $g_\theta \leftarrow g_{\theta 2} + g_{\theta 3}$
      $\theta \leftarrow \theta + ratio \times \alpha \times Adam(g_\theta)$

      **Cooperative process to compress and transcribe via learning signal** $\triangle R$
      $Z = f(x, \theta), \hat{X} = (Z, \eta), \hat{Z} = f(\hat{X}, \theta)$
      $g_{\theta 1} \leftarrow \nabla_{\theta 1} \triangle R(Z(\theta_3), Z(\theta_1, \eta, \theta_2))$
      $g_{\theta 2} \leftarrow \nabla_{\theta 1} \triangle R(Z(\theta_3), Z(\theta_1, \eta, \theta_2))$
      $g_{\theta 3} \leftarrow \nabla_{\theta 3} \triangle R(Z(\theta_3), Z(\theta_1, \eta, \theta_2))$
      $g_\theta \leftarrow g_{\theta 1} + g_{\theta 2} + g_{\theta 3}$
      $g_\eta \leftarrow \nabla_\eta \triangle R(Z(\theta_3), Z(\theta_1, \eta, \theta_2))$
      $\theta \leftarrow \theta - \alpha \times Adam(g_\theta)$
      $\eta \leftarrow \eta - \alpha \times Adam(g_\eta)$
  **end while**

---

Auto-Encoding optimizes the reconstruction quality based on pixel-wise loss, however, CoA-CTRL optimizes the reconstruction quality based on the distribution distance ($\triangle R$) in the feature space, and the feature distribution is defined by the encoder itself, which can be seen as a self-consistent learning strategy in the closed-loop framework. When the encoder tries to find the distribution distance and maximizes it in equation (13), it provides an iterative learning signal ($\triangle R$) for the reconstruction task in equation (14). The encoder takes the responsibility for two roles and learns twice. It minimizes what it maximizes, which is a self-consistent learning strategy (Ma et al., 2022).

In this sense, CoA-CTRL is an adaptive learning strategy and naturally unifies an adversarial process consistent with GAN and a cooperative process consistent with Auto-Encoding, which will contribute to its learning quality and stability. In conclusion, cooperative adversarial learning that follows the value function $\mathcal{T}(\theta, \eta)$ is displayed in equation (15):

$$\min_{\theta, \eta} \max_\theta \mathcal{T}(\theta, \eta) = \triangle R(f(X, \theta), H(X, \theta, \eta)) = \triangle R(Z(\theta), \hat{Z}(\theta, \eta)) \tag{15}$$

### 3.3 Cooperative adversarial ratio (CoA-ratio): a hyperparameter to adjust the cooperative process and the adversarial process

**Active control of the adversarial process and the cooperative process.** As discussed before, the encoder works and learns in two different roles in the adversarial process and the cooperative process respectively. Therefore, through the functioning of the encoder in both processes, we can actively control the adversarial process and the cooperative process. We introduce a hyperparameter cooperative adversarial ratio (CoA-ratio), which is the ratio of the encoder's learning rate in the adversarial process to its learning rate in the cooperative process, to adjust the learning rate of the encoder in two learning processes. In every iteration, before going through equation (13), the learning rate of the encoder would be multiplied by the CoA-ratio, and before going through equation (14), the encoder's learning rate would be restored to the previous value.

**Balance achieved through active control.** The discriminator and generator in GAN work in the opposite direction. The learning process of the discriminator is hard to control, and therefore the balance of the training is hard to keep. Benefited by the encoder learning twice in the cooperative and adversarial processes and its different learning rates and speeds realized by the CoA-ratio, the encoder becomes the controller and regulator of the training process. The training process and

the loss value $\Delta R$ thus can be easily controlled without paying too much attention to constrained network design.

### 3.4 ADVANTAGES AND DIFFERENCES

**Training stability.** Encoder's learning twice provides a way to actively balance the learning processes through its two roles. As equation (15) shows, the encoder not only maximizes $\Delta R$ but also minimizes it, which relieves the need to design special networks or adjust parameters. Compared with former balance techniques, CoA-CTRL does not add constraint techniques or computing processes. It is simple and computationally efficient. In experiments, classic deep nets ResNet18, ResNet50, and ResNet101 (He et al., 2016a;b) are used to validate CoA-CTRL's active balance.

**Sample-consistent reconstruction.** As mentioned in 3.2, CoA-CTRL naturally unifies the learning strategies of Auto-Encoding and GAN, which will help the encoder learn better and faster. Other than that, it will benefit the sample-wise reconstruction. Sample-wise consistent reconstruction $g(f(x)) \approx x$ is the ideal solution to $\Delta R(Z(\theta), Z(\theta, \eta)) \approx 0$, and $\Delta R(Z(\theta), Z(\theta, \eta))$ is determined by encoder $f(x, \theta)$ and decoder $g(z, \eta)$. CTRL (Dai et al., 2022) would minimize $\Delta R(\theta, \eta)$ merely through the decoder $g(z, \eta)$, which would give an approximate optimization choice that results in poor sample-wise consistency. However, if we optimize $\Delta R(\theta, \eta)$ by decoder $g(z, \eta)$ and encoder $f(x, \theta)$, the optimization process would become simple, and the ideal sample-consistent solution would be easily obtained. In addition, cooperative adversarial learning via closed-loop transcription produces good disentangled feature space. Later experiments will demonstrate this advantage.

**Simpler.** As Auto-Encoding, its variants (Kingma & Welling, 2013), and GAN all gain a lot of attention in the generative model area, many works have attempted to combine Auto-Encoding and GAN, like Bigan (Donahue et al., 2016), ALI (Dumoulin et al., 2016), adversarial autoencoders (Makhzani et al., 2015) and VAE-GAN (Larsen et al., 2016). Different from those attempts, CoA-CTRL implements the closed-loop transcription (Dai et al., 2022), introduces the cooperative process, and invites no other discriminator. The encoder learns twice in different roles within one iteration without investing more computing resources. Stability and balance are controlled actively without regulation techniques, which contributes to computing resource saving compared to other regulation techniques like spectral normalization (Miyato et al., 2018). Also, the original CTRL always depends on big batch sizes, while cooperative adversarial learning could reduce this demand.

## 4 EXPERIMENTS

### 4.1 SETTING

In this paper, we intend to justify two main advantages of CoA-CTRL: firstly, CoA-CTRL's robustness to different net architectures through our cooperative adversarial learning; secondly, CoA-CTRL's sample-consistent reconstruction. To conduct the experiments, we use two types of encoders: deep encoders and normal encoders, and both types of encoders will be paired with one type of decoder. We conduct the experiments with deep encoders on the diverse data set CIFAR-10 (Krizhevsky et al., 2009) and STL-10 (Coates et al., 2011), as well as the facial data set Celeb-A (Liu et al., 2015), aiming to demonstrate CoA-CTRL's active balance. We conduct the experiments with normal encoders on MNIST (LeCun et al., 1998), CIFAR-10, and ImageNet-1k (Russakovsky et al., 2015), aiming to prove CoA-CTRL's sample-wise consistency. More details of the experiment setting could be found in Appendix A.2.

### 4.2 EMPIRICAL VERIFICATION OF COA-CTRL'S ACTIVE BALANCE

#### 4.2.1 ACTIVE BALANCE TO NET ARCHITECTURES

To verify CoA-CTRL's active balance, we conduct several comparative tests on CIFAR-10, using ResNet18, ResNet50, and ResNet101 as the encoder, and the widely used 8-layer resnet (De8) (Miyato et al., 2018) as the decoder. We intend to prove that even with an unbalanced combination of the encoder and decoder, CoA-CTRL can perform well in a stable manner. We apply the same settings on GAN and CTRL, aiming to compare their stability and performances with CoA-CTRL. Results in Table 1 show that CoA-CTRL works well, while GAN and CTRL fail and collapse in the training process.

To quantity CoA-CTRL's performance, we test CoA-CTRL by the widely used Inception score (IS) (Salimans et al., 2016) and Fréchet Inception Distance (FID) (Heusel et al., 2017). We further compare CoA-CTRL's IS and FID with other major generative models, which are displayed in Table 2.

Interested in the parts feature dimension (nz) and batchsize (bs) play in CoA-CTRL's performance and stability, we additionally adjust nz and batchsize to see whether CoA-CTRL can maintain balance. As shown in Table 1, we find that CoA-CTRL works well with different combinations of nz and batchsize, and it performs better when we increase nz and batchsize at the same time.

We also explore the loss value $\Delta R$ in the training process. As shown in Figure 3, CoA-CTRL keeps low and stable $\Delta R$ and high $R(Z \cup \hat{Z})$, while CTRL shows unstable training loss $\Delta R$. CoA-CTRL's stable training loss contributes to its training success and excellent performance.

Table 1: Stability and performance of CoA-CTRL compared with GAN and CTRL on CIFAR-10. Experiments show that CoA-CTRL gets excellent performances in a stable manner even with unbalanced settings of a deep encoder and a shallow decoder. Avg. of $R$ and Avg. of $\Delta R$ refer to the average value of $R(Z \cup \hat{Z})$ and $\Delta R$ in the training process. $\uparrow$ means higher is better, and $\downarrow$ means lower is better.

| Methods | Encoder/ Discriminator | Decoder/ Generator | nz | bs | Result | Avg. of $R$ | Avg. of $\Delta R$ | IS $\uparrow$ | FID $\downarrow$ |
|---|---|---|---|---|---|---|---|---|---|
| **(1) Comparisons of stability and performance** | | | | | | | | | |
| GAN | ResNet18 | De8 | 128 | 128 | fail | - | - | - | - |
| GAN | ResNet18 | De8 | 128 | 512 | fail | - | - | - | - |
| CTRL | ResNet18 | De8 | 128 | 512 | fail | 45.57 | 9.77 | - | - |
| CoA-CTRL | ResNet18 | De8 | 128 | 512 | succeed | 65.03 | 1.65 | 7.94 | 10.49 |
| CoA-CTRL | ResNet50 | De8 | 128 | 512 | succeed | 64.04 | 1.84 | 7.82 | 11.17 |
| CoA-CTRL | ResNet101 | De8 | 128 | 512 | succeed | 63.79 | 1.76 | 7.12 | 19.12 |
| **(2) Ablation study on nz and batchsize** | | | | | | | | | |
| CoA-CTRL | ResNet18 | De8 | 128 | 512 | succeed | 65.03 | 1.65 | 7.94 | 10.49 |
| CoA-CTRL | ResNet18 | De8 | 128 | 1024 | succeed | 63.85 | 1.25 | 7.91 | 10.92 |
| CoA-CTRL | ResNet18 | De8 | 256 | 512 | succeed | 127.92 | 6.15 | 7.38 | 12.72 |
| CoA-CTRL | ResNet18 | De8 | 256 | 1024 | succeed | 127.15 | 2.81 | **8.21** | **9.54** |
| CoA-CTRL | ResNet18 | De8 | 512 | 1024 | succeed | 244.38 | 11.17 | 7.60 | 11.61 |

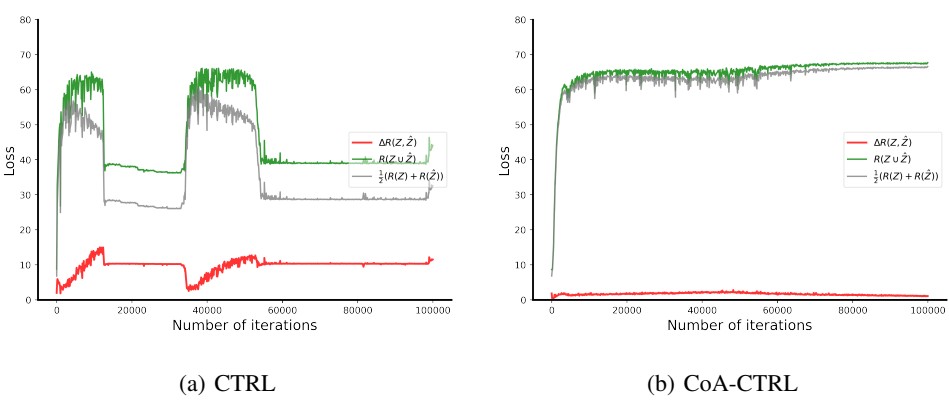

(a) CTRL        (b) CoA-CTRL

Figure 3: Loss evaluation of CTRL and CoA-CTRL in the training process on CIFAR-10, using ResNet18. CoA-CTRL keeps $\Delta R$ in a stable curve even with an unbalanced setting of a deep encoder and a shallow decoder.

### 4.2.2 ASSOCIATION BETWEEN THE LOSS VALUE $\Delta R$ AND PERFORMANCE

As shown in Table 1 and Figure 3, CoA-CTRL's performance seems to be associated with $\Delta R$. As discussed in section 3.3, we introduce CoA-ratio to adjust the learning rate of the encoder in the adversarial process and the cooperative process, which would influence the loss value $\Delta R$. Therefore, we set CoA-catio at 1.25,1.5,1.75 and 2.0 respectively to explore the influence of $\Delta R$ on CoA-

Table 2: Comparison of performances of CoA-CTRL and other generative methods on CIFAR-10 and STL-10. Data of other generative models are cited from relevant papers. For CoA-CTRL (1), the nz is 128, and the bs is 512; for CoA-CTRL (2), the nz is 256, and the bs is 1024.

| Methods | CIFAR-10 | | STL-10 | |
|---|---|---|---|---|
| | IS↑ | FID↓ | IS↑ | FID↓ |
| SNGAN | 8.2 | 21.7 | 9.1 | 40.1 |
| WGAN-GP | 7.9 | 29.3 | - | - |
| WGAN-ALP | 8.3 | 13.0 | - | - |
| SSGAN | - | 17.1 | - | - |
| LOGAN | 8.7 | 17.7 | - | - |
| BIGGAN | 9.2 | 14.7 | - | - |
| DCVAE | 8.2 | 17.9 | 8.1 | 41.9 |
| CTRL | 8.1 | 19.6 | 8.4 | 38.6 |
| **CoA-CTRL (1)** | 7.94 | **10.49** | 8.68 | **22.96** |
| **CoA-CTRL (2)** | 8.21 | **9.54** | 9.28 | **18.54** |

CTRL's performance. We conduct the experiment on CIFAR-10, using ResNet18 as the encoder, with nz and batchsize set at 128 and 512. As shown in Table 12 and Figure 6 in the Appendix, different CoA-ratios are associated with different values of $\Delta R$ and $R$, and the performance score IS and FID are highly influenced by $\Delta R$, but the good thing is that the change of $\Delta R$ is stable. This paper just points out that CoA-ratio, $\Delta R$, and performance are highly associated. Future works will explore the mechanisms.

### 4.2.3 EXCELLENT PERFORMANCE ACHIEVED THROUGH A SIMPLE DECODER

We compare CoA-CTRL's performance with other generative models on CIFAR-10 and STL-10. The data in Table 2 is directly cited from those relevant papers except for CoA-CTRL. For CoA-CTRL, We use ResNet18 as the encoder on both CIFAR-10 and STL-10. Table 2 shows that CoA-CTRL performs better with a simple decoder and experiment setting, compared with other generative models, such as GANs with regularization techniques (SNGAN, WGAN-GP, WGAN-ALP) (Miyato et al., 2018; Gulrajani et al., 2017; Terjék, 2019), self-supervised GAN (Chen et al., 2019), latent optimisation GAN (LOGAN) (Wu et al., 2019), complex model GAN (BIGGAN) (Brock et al., 2018), a recent combination of GAN and VAE (DCVAE) (Parmar et al., 2021), and the original CTRL (Dai et al., 2022). Compared with CTRL, the FID value of CoA-CTRL is decreased by 10.06 on CIFAR-10 and 20.06 on STL-10. The improvements are clear and substantial.

### 4.3 SAMPLE-WISE CONSISTENCY

CoA-CTRL performs well on sample-wise reconstruction in terms of sample features, which is demonstrated by our experiments on several mainstream data sets using normal encoders. Figure 4 shows CoA-CTRL's reconstruction performance on MNIST compared with CTRL. We can see that CoA-CTRL's reconstruction is almost the same as the original input, better and more consistent than CTRL. For CIFAR-10 and ImageNet-1k (Russakovsky et al., 2015), we use networks listed through Table 5 to Table 9 in the Appendix. We run 20,000 iterations on both data sets. Figure 4 displays CoA-CTRL's performance on CIFAR-10 and ImageNet-1k. We can see that CoA-CTRL reconstructs well in terms of features, color, and classes, which is benefited from cooperative adversarial learning and a loss function based on the feature space.

### 4.4 DISENTANGLED FEATURE SPACE

The latent space of GAN has no certain meanings and lacks inverse maps from data to the latent space. Some following works discussed this issue (Chen et al., 2016; Karras et al., 2019; 2020; Tov et al., 2021). The latent space in CoA-CTRL has clear and disentangled meanings. CoA-CTRL possesses the concept of dual consistent maps, $x \rightarrow z$, and $z \rightarrow x$. Images in Figure 8 in the Appendix are the generated samples of CIFAR-10 along independent principal components. We select the top 10 components with every row referring to a component from top to bottom. We can see that different shapes, styles, backgrounds, and other visual attributes are well modeled in different principal components, and the images vary with the scale value. In addition, we test

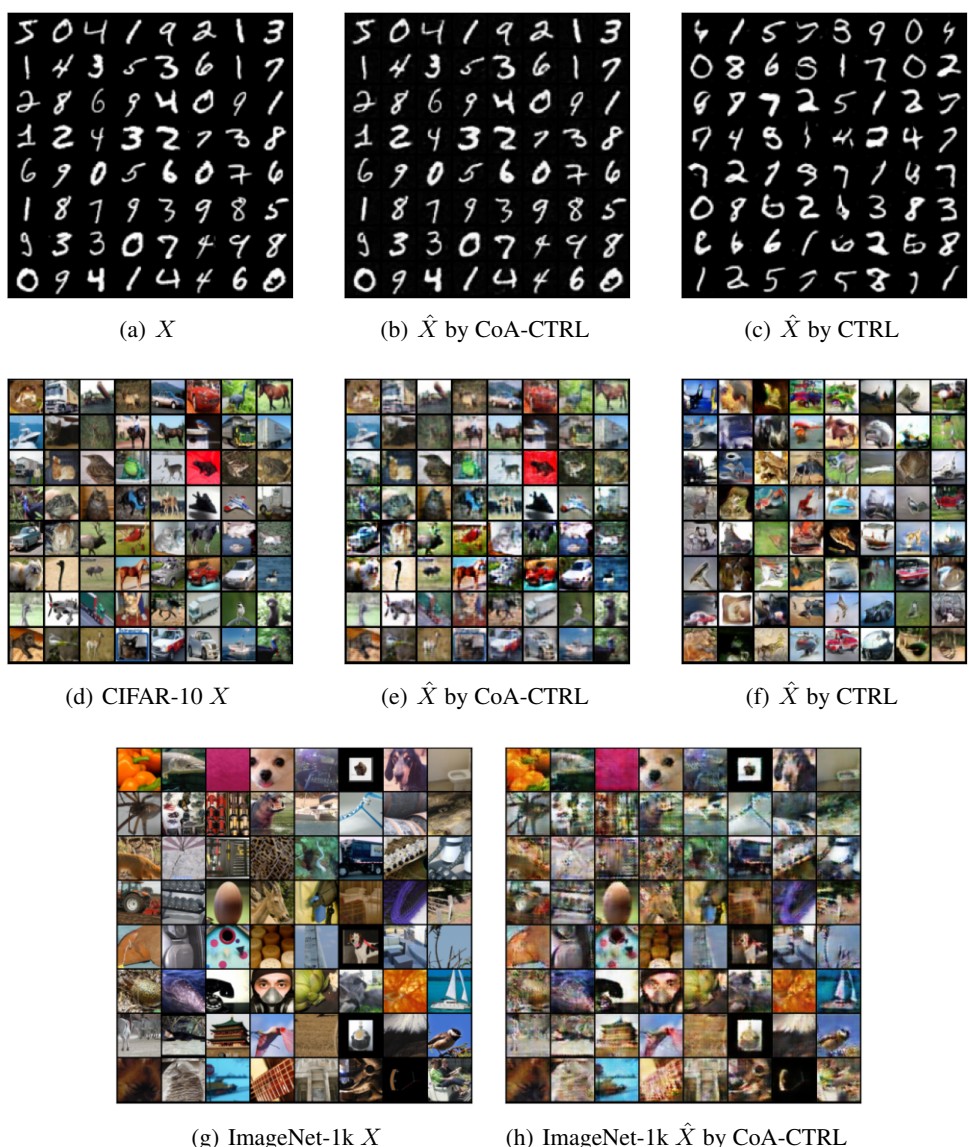

Figure 4: Comparison of sample-wise reconstruction. CoA-CTRL performs well in sample-wise reconstruction on MNIST, CIFAR-10, and ImageNet-1k.

the classification accuracy on MNIST based on feature representations with one Nearest Neighbor (1NN) classifier. As Table 13 in the Appendix shows, CoA-CTRL's performance is competitive compared with other methods (Springenberg, 2015; Kingma & Welling, 2013; Donahue et al., 2016; Dumoulin et al., 2016; Makhzani et al., 2015; Parmar et al., 2021; Dai et al., 2022).

## 5 DISCUSSION AND CONCLUSION

In this paper, we propose cooperative adversarial learning, and based on this new learning method and closed-loop transcription, we build a promising generative model, which possesses the properties of active balance, better generative performance, and disentangled latent space. Other than that, we find it competitive in unsupervised representation. Although cooperative adversarial learning provides a way to balance deep nets, some questions are still unclear. For example, whether a deeper encoder would benefit to better performance, and how big a coop-ratio or $\Delta R$ is best for the training process and model performance. These questions deserve further explorations.

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

# A APPENDIX

## A.1 LEARNING STRATEGY OF THE ORIGINAL CTRL

---
**Algorithm 2** Original CTRL's learning strategy

---
**Require:** $\alpha$, learning rate. $ratio$, CoA-ratio. $\epsilon^2$, coding rate parameter. $bs$, batch size.
**Require:** $\theta$, init parameters of encoder. $\eta$, init parameters of decoder.
  **while** $\eta$ has not converged **do**

     Sample $X \leftarrow x^{(i)}{}_{i=1}^{bs}$ a batch from the real data
     $Z = f(x, \theta), \hat{X} = (Z, \eta), \hat{Z} = f(\hat{X}, \theta)$
     $g_{\theta 2} \leftarrow \nabla_{\theta 2} \Delta R(Z(\theta_3), \hat{Z}(\theta_1, \eta, \theta_2))$
     $g_{\theta 3} \leftarrow \nabla_{\theta 3} \Delta R(Z(\theta_3), \hat{Z}(\theta_1, \eta, \theta_2))$
     $g_\theta \leftarrow g_{\theta 2} + g_{\theta 3}$
     $\theta \leftarrow \theta + ratio \times \alpha \times Adam(g_\theta)$

     $Z = f(x, \theta), \hat{X} = (Z, \eta), \hat{Z} = f(\hat{X}, \theta)$
     $g_\eta \leftarrow \nabla_\eta \Delta R(Z(\theta_3), Z(\theta_1, \eta, \theta_2))$
     $\eta \leftarrow \eta - \alpha \times Adam(g_\eta)$
  **end while**

---

## A.2 EXPERIMENT SETTING

### A.2.1 EXPERIMENTS USING NORMAL ENCODERS

We conduct the experiments using nets in DCGAN (Radford et al., 2015) and some other simple nets on MNIST, CIFAR-10, and ImageNet-1k, aiming to justify CoA-CTRL's sample-wise consistency. The details of the networks can be found through Table 3 to Table 7. The experiments we conduct are fair comparisons to the original CTRL, and the only difference is the learning strategy. For experiments with normal encoders, the encoder and the decoder have similar volumes. We set the hyperparameters $\beta_1$ and $\beta_2$ of the optimizer Adam (Kingma & Ba, 2014) at 0.0 and 0.9 for MNIST, and at 0.5 and 0.9 for CIFAR-10 and ImageNet-1k. We set the learning rate at 0.0001 and apply linear decay. $\epsilon^2$ is set at 0.5. We adjust the CoA-ratio at 1.3 for MNIST and 1.5 for other data sets. For MNIST, we set batchsize at 256 and run 10,000 iterations. For CIFAR-10, we set the batchsize at 512 and run 20,000 iterations. For ImageNet-1k, we set the batchsize at 128 and run 20,000 iterations.

### A.2.2 EXPERIMENTS USING DEEP ENCODERS

We conduct experiments using deep encoders on CIFAR-10, STL-10, and Celeb-A. The settings of the experiments using deep encoders are as follows. Adam (Kingma & Ba, 2014) would be used as the optimizer. The learning rate is set at 0.0001, and the linear decay is applied. For the classic hyperparameters $\beta_1$ and $\beta_2$, we set them at 0.0 and 0.9 respectively. We fix $\epsilon^2$ at 0.5 in all experiments. CoA-ratio is set at 1.5. The value of nz is set at 128. The batchsize is 512. For the decoder, we adopt the widely used networks in DCGAN (Radford et al., 2015), SNGAN (Miyato et al., 2018), and CTRL (Dai et al., 2022). The details can be found in Table 5, Table 9, and Table 10. For the encoder, we apply deep nets 18-layer preaction resnet (ResNet18), 50-layer preaction resnet (ResNet50), and 101-layer preaction resnet (ResNet101) to verify CoA-CTRL's stability and robustness to deep nets.

As for ResNet18, ResNet50, and ResNet101 in this paper, we use preaction (He et al., 2016b) and average pooling to downsample, which will contribute to better feature extraction. For STL-10 and Celeb-A, we add a downsample at the first ResBlock of ResNet18. Spectral normalization (Miyato et al., 2018), batch normalization (Ioffe & Szegedy, 2015), or other regulation techniques are not applied, instead, just a simple and standard convolution layer without constraint is employed.

We run 10,000 iterations on MNIST, 100,000 iterations on CIFAR-10, 150,000 iterations on STL-10 and Celeb-A. We resize the resolution of MNIST to $32 \times 32$, STL-10 to $48 \times 48$, and Celeb-A to $64 \times 64$.

Table 3: Decoder for MNIST

| $z \in \mathbb{R}^{dim}$ |
| --- |
| 4×4,stride=1,pad=0 deconv.BN256 ReLU |
| 4×4,stride=2,pad=1 deconv.BN128 ReLU |
| 4×4,stride=2,pad=1 deconv.BN64 ReLU |
| 4×4,stride=2,pad=1 deconv 1 Tanh |

Table 4: Encoder for MNIST

| $x \in \mathbb{R}^{32 \times 32 \times 1}$ |
| --- |
| $4 \times 4$, stride=2, pad=1 conv 64 lReLU |
| $4 \times 4$, stride=2, pad=1 conv. BN 128 lReLU |
| $4 \times 4$, stride=2, pad=1 conv. BN 256 lReLU |
| $4 \times 4$, stride=1, pad=0 conv 128 |

Table 5: Decoder for CIFAR-10

| $z \in \mathbb{R}^{dim}$ |
| --- |
| dense $\rightarrow 4 \times 4 \times$ 256 |
| ResBlock up 256 |
| ResBlock up 256 |
| ResBlock up 256 |
| BN, ReLU, 3×3 conv, 3 Tanh |

Table 6: Encoder for CIFAR-10

| $x \in \mathbb{R}^{32 \times 32 \times 1}$ |
| --- |
| ResBlock down 64 |
| ResBlock down 128 |
| ResBlock down 256 |
| $4 \times 4$, stride=1, pad=0 conv 512 |

Table 7: Decoder for ImageNet

| $z \in \mathbb{R}^{dim}$ |
| --- |
| 4×4,stride=1,pad=0 deconv.BN512 ReLU |
| 4×4,stride=2,pad=1 deconv.BN256 ReLU |
| 4×4,stride=2,pad=1 deconv.BN128 ReLU |
| 4×4,stride=2,pad=1 deconv.BN64 ReLU |
| 4×4,stride=2,pad=1 deconv 3 Tanh |

Table 8: Encoder for ImageNet

| $x \in \mathbb{R}^{64 \times 64 \times 3}$ |
| --- |
| $4 \times 4$, stride=2, pad=1 conv 64 lReLU |
| $4 \times 4$, stride=2, pad=1 conv. BN 128 lReLU |
| $4 \times 4$, stride=2, pad=1 conv. BN 256 lReLU |
| $4 \times 4$, stride=2, pad=1 conv 512 lReLU |
| $4 \times 4$, stride=1, pad=0 conv 1024 |

Table 9: Decoder for Celeb-A

| $x \in \mathbb{R}^{64 \times 64 \times 3}$ |
| --- |
| ResBlock down 64 |
| ResBlock down 128 |
| ResBlock down 256 |
| ResBlock down 512 |
| $4 \times 4$, stride=1, pad=0 conv 1024 |

Table 10: Decoder for STL-10

| $z \in \mathbb{R}^{dim}$ |
| --- |
| dense $\rightarrow 6 \times 6 \times$ 512 |
| ResBlock up 256 |
| ResBlock up 128 |
| ResBlock up 64 |
| BN, ReLU, 3×3 conv, 3 Tanh |

### A.2.3 STRUCTURES OF RESBLOCK

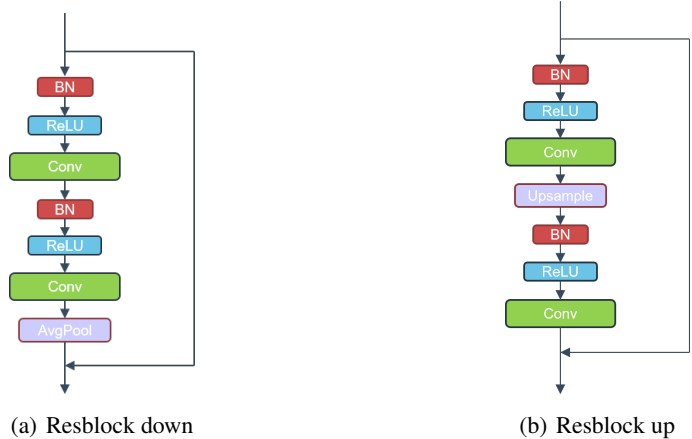

(a) Resblock down  (b) Resblock up

Figure 5: Structure of resblock down and reblock up.

### A.3 CoA-CTRL'S ROBUSTNESS TO BATCH-SIZE

Table 11: Comparison between CoA-CTRL and the original CTRL in terms of robustness to batch size, using normal encoder and decoder as displayed in Table 5 and Table 6. Fail means the model collapses.

| Batch size | 64 | 128 | 256 | 512 | 1024 |
|---|---|---|---|---|---|
| CTRL | Fail | Fail | Fail | succeed | succeed |
| CoA-CTRL | succeed | succeed | succeed | succeed | succeed |

### A.4 ABLATION STUDY ON CoA-RATIO

Table 12: Ablation study on CoA-ratio on CIFAR-10. Avg. of $R$ and Avg. of $\Delta R$ refer to the average value of $R(Z \cup \hat{Z})$ and $\Delta R$ in the training process.

| CoA-ratio | 1.25 | 1.5 | 1.75 | 2.0 |
|---|---|---|---|---|
| Avg. of $R$ | 64.08 | 65.03 | 63.45 | 64.16 |
| Avg. of $\Delta R$ | 0.62 | 1.65 | 3.92 | 4.07 |
| IS↑ | 6.55 | 7.94 | 7.02 | 6.64 |
| FID↓ | 22.34 | 10.49 | 38.04 | 38.20 |

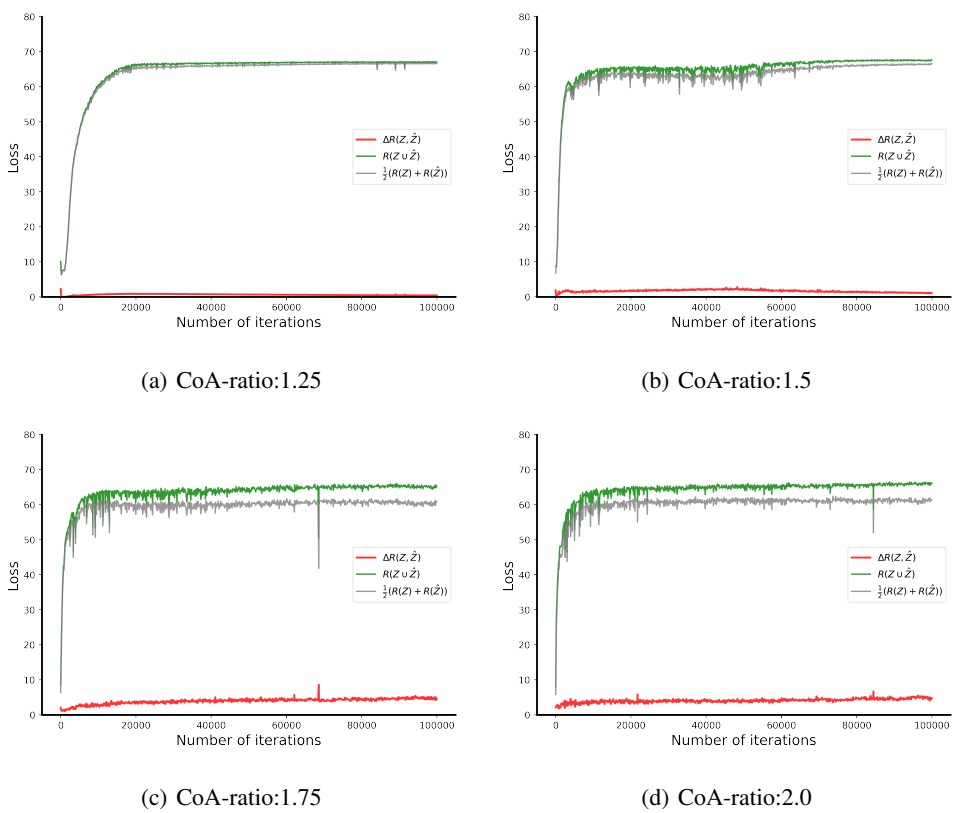

Figure 6: Loss evaluation of CoA-CTRL in the training process on CIFAR-10, using ResNet18 as the encoder and De8 as the decoder. Different CoA-ratio of 1.25, 1.5, 1.75, and 2.0 are used.

## A.5 COA-CTRL'S SAMPLE-WISE RECONSTRUCTION ON STL-10 AND CELEB-A

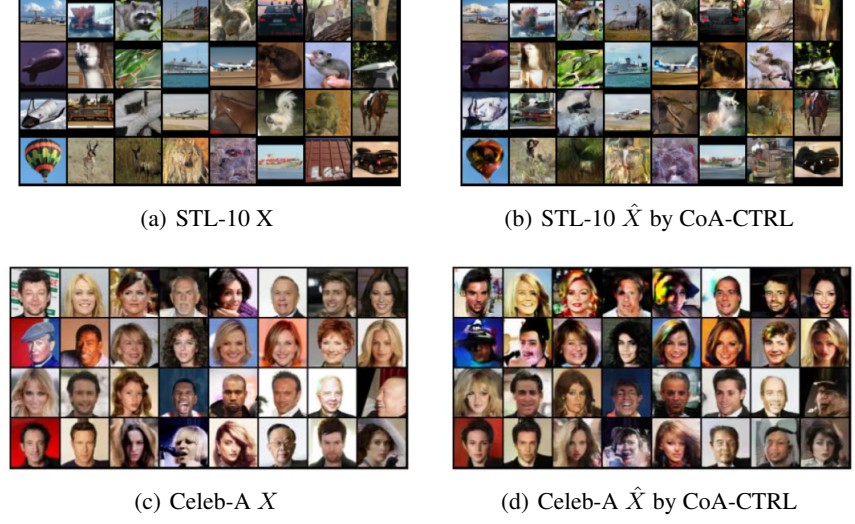

Figure 7: Comparison of sample-wise reconstruction. CoA-CTRL performs good sample-wise reconstruction STL-10 and Celeb-A.

A.6   DISENTANGLED FEATURE SPACE AND CLASS-WISE ACCURACY

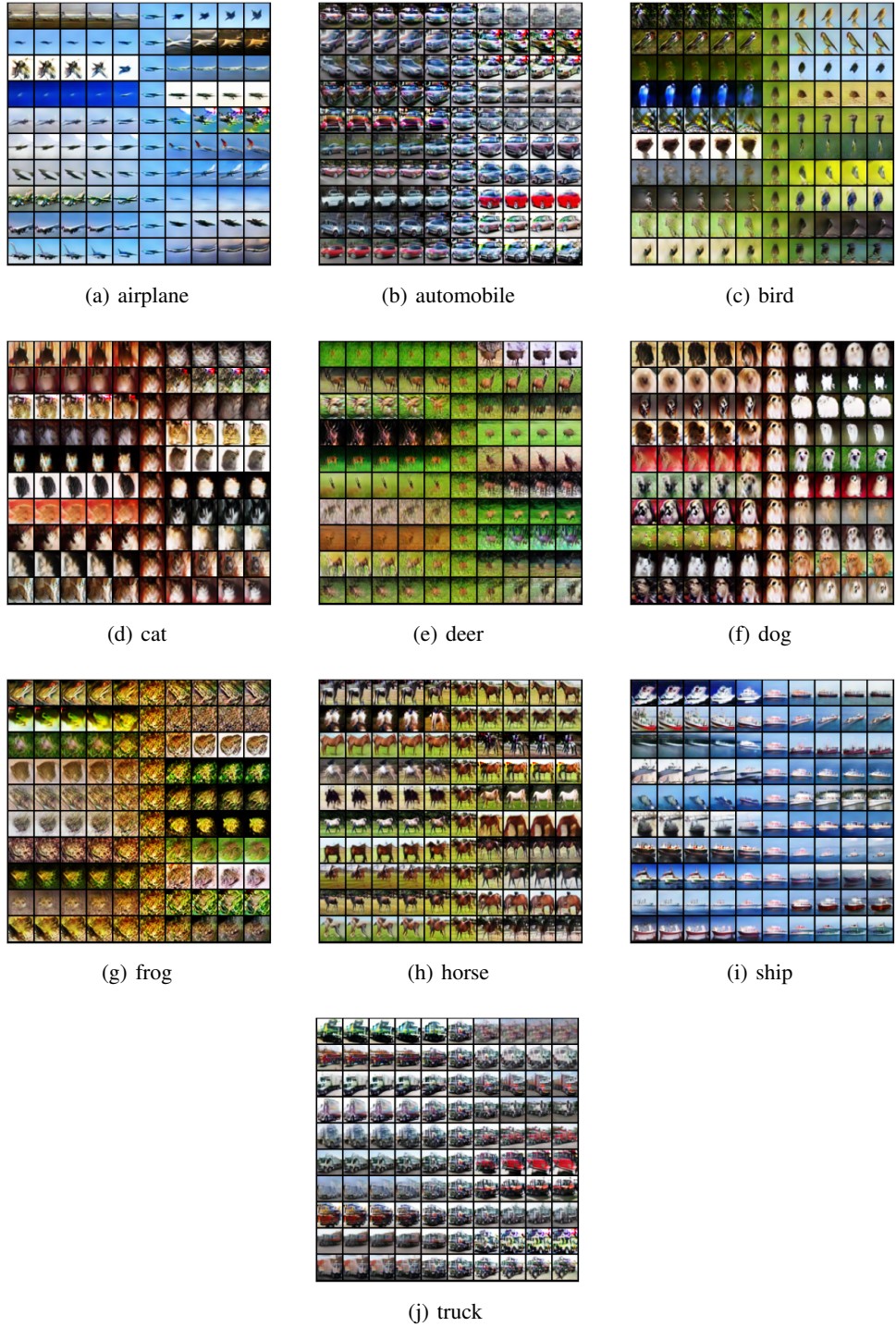

Figure 8: Generated samples of ten classes along 10 principal components from the feature subspace on CIFAR-10. Every row refers to a principal component, and the scale value varies from -1.5 to 1.5.

Table 13: Class-wise accuracy performance with respect to unsupervised representation on MNIST

| Method | VAE | BIGAN | catGAN | AAE | DC-VAE | CTRL | CoA-CTRL |
|---|---|---|---|---|---|---|---|
| Accuracy | 97.12% | 97.39% | 95.7% | 95.9% | 98.71% | 89.12% | 95.89% |

