# OpenReview forum: "Cooperative Adversarial Learning via Closed-Loop Transcription"
_ICLR.cc/2023/Conference — Submitted to ICLR 2023_

### Official Review · Reviewer_AoR2 · 2022-10-24

**Confidence:** 3
**Correctness:** 3
**Technical Novelty And Significance:** 3
**Empirical Novelty And Significance:** 2
**Recommendation:** 5

**Clarity, Quality, Novelty And Reproducibility:**

- Clarity: The paper is well-written
 - Quality: Overall, I think this paper is clear and easy to understand.
 - Novelty: I think the biggest novelty is the use of the integration of cooperative adversarial
learning and closed-loop transcription. More experiments regarding this part need to be performed (See weaknesses).
 - Reproducibility: This paper does not release the code in the current version.


**Strength And Weaknesses:**

Pros

- This paper proposed a novel generative model with cooperative adversarial
learning and closed-loop transcription. The idea of integrating cooperative adversarial
learning and CTRL to improve the robustness of generative models is interesting.
- The experiments evaluate the robustness of the model on some datasets.

Cons

- Previous works [1,2] have already improved the robustness of the generative models, can you explain the advantage of the CoA-CTRL with these previous works?
- The datasets used in this paper are somewhat small in comparison. Just wondering about the performance of CoA-CTRL on CIFAR-100 and ImageNet

[1] Regularizing Generative Adversarial Networks under Limited Data

[2] Training generative adversarial networks with limited data

**Summary Of The Paper:**

This paper proposed a novel generative model (CoA-CTRL) that integrates cooperative adversarial
learning and closed-loop transcription. Experiments on MNIST, CIFAR-10, STL-10, and Celeb-A demonstrate the robustness and consistency of the proposed model.

**Summary Of The Review:**

This paper has some interesting points, but this version lacks a deeper explanation of the proposed framework. Some theoretical or experimental explanations are encouraged to improve the paper. Also, it lacks comparison to some existing works, this paper needs to compare their performance on a better-implied baseline. Thus I give 5 at the current time.

---

> ### Author Response · Authors · 2022-11-19
> **Responses to Reviewer AoR2**
>
> We thank you for your encouraging review. Here are our responses.
>
> Q1:
>
> Most of previous works focus on regulation directions for robustness. Cooperative adversarial learning focuses on learning and optimization strategy.
>
> 1 We do not use additional resources to improve robustness.
>
> 2 Instability of the training process may come from unstable learning signals. Previous works tend to limit the discriminator to provide learning signals, while cooperative adversarial learning provides stable learning signals by cooperative methods inside of system.
>
>
>
> Q2:
>
> We have provided experiment results of ImageNet-1k in the revised paper.

---

### Official Review · Reviewer_qmAD · 2022-10-25

**Confidence:** 5
**Clarity, Quality, Novelty And Reproducibility:** see above.
**Correctness:** 3
**Technical Novelty And Significance:** 2
**Empirical Novelty And Significance:** 1
**Recommendation:** 1

**Strength And Weaknesses:**

This paper has already got published in Entropy 2022 [1]. Why is it submitted here?

[1] https://www.mdpi.com/1099-4300/24/4/456

**Summary Of The Paper:**

This paper tries to combine the autoencoding (cooperative learning) and GAN (adversarial learning) in a unified framework such that it proposes to apply the encoder to play two different roles and control different objectives.

**Summary Of The Review:**

see above.

---

> ### Author Response · Authors · 2022-11-19
> **Responses to Reviewer qmAD**
>
> We are sorry for that let you misunderstanding our paper
>
> CTRL provide a framework,
> Cooperative aversararial learning provide a learning strategy for CTRL
>
> You can take a look at my revised article specially for chapter 3.2

---

### Official Review · Reviewer_1daZ · 2022-10-25

**Confidence:** 3
**Correctness:** 1
**Technical Novelty And Significance:** 1
**Empirical Novelty And Significance:** 1
**Recommendation:** 3

**Clarity, Quality, Novelty And Reproducibility:**

Overall, the novelty of the work is marginal since it is a simple modification to the existing method (Dai et al., 2022), where the correctness of the modification is in doubt. Some figures are also largely from prior works (Dai et al., 2022 and Chan et al., 2022).

**Strength And Weaknesses:**

Strength:
- The idea presented in the paper is clear.
- The paper is easy-to-read in general and reasonably self-contained.


Weakness:
- Based on my understanding, the adversarial and cooperate optimization in Eq.(9) and Eq.(10), respectively, give exactly the same gradient with respect to the encoder parameters but in the opposite direction (i.e., only the sign of the gradient is different) since both optimizations are with respect to the same decoder parameters \gamma. Considering that the authors proposed to apply different learning rates for the encoder when optimizing Eq.(9) and Eq.(10) (i.e., named CoA-ratio), these two optimizations eventually collapse to the same adversarial optimization in Dai et al., 2022 but applying different learning rates for encoder and decoder. Please clarify it in the rebuttal.

- The experiments are generally not convincing. As discussed in Section 4.2.1, the authors intentionally employed unbalanced encoder and decoder to make training unstable. As a result, all baselines such as GAN and CTRL fail to learn even an extremely simple dataset like CIFAR10. This is an unreasonable result considering the advances in the field and the low complexity of the dataset. The authors instead should present the main comparison results with the balanced setting and use Table 1 as an additional analysis.

- Figure 5 is supposed to be evidence to show that the proposed method learns disentangled representation, yet the results display the opposite. As it shows, changing one factor of variation (each row in Figure 5) leads to changes in multiple attributes, which show evidence of entangled representation.


**Summary Of The Paper:**

This paper proposed to extend the Closed-Loop Transcription (CTRL) (Dai et al., 2022). While the original CTRL formulates the task of generative modeling as a minimax game between the encoder and the decoder, the proposed method simply employs an additional cooperate term that minimizes the rate reduction in both of the players. The proposed method is evaluated with several GAN and VAE-GAN baselines including CTRL and demonstrated some improvements on simple datasets like CIFAR10 and MNIST.

**Summary Of The Review:**

I lean toward rejecting this paper due to non-trivial flaws in the method and experiments.

---

> ### Author Response · Authors · 2022-11-19
> **Responses to Reviewer 1daZ**
>
> \usepackage[colorlinks,linkcolor=red]{hyperref}
>
> Q1:Does cooperative adversarial leaning just simply change the direction of encoder in cooperative process and adversarial process and value is the same？
>
> A1:
> Answer is No. Not only the sign is different, but also the value is different. Closed-loop transcription uses the encoder twice, which is different from GAN. In closed-loop transcription, the encoder will be calculated the gradient three times in the backward propagation process. So, the gradient value of encoder in the adversarial process is last gradient values, which is consistent with GAN, but in the cooperative process, we calculate all gradient values in the closed-loop transcription. This is the most important part of cooperative adversarial learning.
> We have added the algorithm of cooperative adversarial learning in the revised paper. Details can be found in 3.2 as well as Algorithm 1 in the revised paper.
>
> Q2:
>
> A2:please refer to Clarification of motivation
>
>
> Q3:
>
> A3:disentangled  representation have not explanation， what i want to express is that different visual features correspond to different principal components (every rows refer to a principal components)

---

### Official Review · Reviewer_rrv8 · 2022-10-27

**Confidence:** 3
**Clarity, Quality, Novelty And Reproducibility:** The paper is not clearly written and …
**Correctness:** 2
**Technical Novelty And Significance:** 2
**Empirical Novelty And Significance:** 2
**Recommendation:** 3

**Strength And Weaknesses:**

Pros:
+ The idea of cooperative adversarial learning is an important and interesting direction.

Cons:
- The paper does not clearly explain the motivations and the ideas.
- The details of the proposed framework are not illustrated in a clear way.
- The experimental results do not fully support the conclusion of the paper.



**Summary Of The Paper:**

The paper designs a generative model for effective cooperative adversarial learning through closed-loop transcription.  In the training, both the encoder and decoder are trained simultaneously for cooperative adversarial learning. The experimental results show effectiveness of the proposed strategy.

**Summary Of The Review:**

The techinical novelty is below the bar of ICLR.

---

> ### Author Response · Authors · 2022-11-19
> **Response to Reviewer rrv8**
>
> We thank you for your valuable opinions. For your questions, we have addressed some in the general comments, and we want to address some further questions in this part.
>
> Q1:The paper does not clearly explain the motivations and the ideas.
>
> A1:Original CTRL unifies the structure of Auto-Encoding and GAN in its framework, but it just follows the learning strategy of GAN, instead of both. Besides, the original CTRL has certain problems. For example, it cannot achieve sample-wise consistency like Auto-Encoding, and its training process is unstable. We want to provide a learning strategy that fits CTRL and improves its sample-wise consistency, training stability, and overall performance.
>
> Q2:The details of the proposed framework are not illustrated in a clear way.
>
> A2:We have modified our paper. You can see the modifications which are marked in red in the revised paper.
>
>
> Q3:The experimental results do not fully support the conclusion of the paper.
>
> A3:We have added some experiments in the revised paper. The relevant parts have been marked red.

---

> > ### Comment · Reviewer_1daZ · 2022-11-28
> > **Thanks for the response.**
> >
> > I appreciate the authors' response. I read the rebuttal and other reviews carefully. Unfortunately, I am not convinced about the authors' response on A2 and A3, and have concerns about the general presentation of the paper.
> >
> > Regarding A1, although I appreciate the revised manuscript and algorithm, it is still quite unclear how exactly the gradient of the adversarial and cooperative processes are computed due to (1) unclear/abused notations and (2) confusing text descriptions. For instance, the authors used \theta_1 to \theta_3 to differentiate the encoder gradients, but they are all essentially \theta while computed with respect to different input/output. Also, in Algorithm 1, it is difficult to understand the difference between the gradient with respect to \theta_1 and \theta_2 in the cooperative process since they are essentially the same equations. Overall, I can see that the encoder gradients in adversarial and cooperative processes have different values since the latter is computed with respect to the updated parameters by the former, but nothing more than that.
> >
> > Regarding A2 and A3, I am not convinced by the author's responses. The experiments are still biased and do not reflect the common observations from the literature.
> >
> > Overall, I believe that the paper needs significant improvement in presentation and experiments to meet ICLR standards. Hence, I maintain my recommendation to rejection.

---

> > > ### Author Response · Authors · 2022-11-29
> > > **Thanks for your kind response.**
> > >
> > > Thanks for your time and suggestions. We will further improve our presentation and expression based on the issues you point out.

---

### Official Review · Reviewer_kFbE · 2022-10-31

**Confidence:** 3
**Correctness:** 2
**Technical Novelty And Significance:** 2
**Empirical Novelty And Significance:** 2
**Recommendation:** 5

**Clarity, Quality, Novelty And Reproducibility:**

The paper is in general well written and easy to follow. However, some description in the equation part are a bit confusing. More clarification may be needed for readers who are not familiar with $MCR^2$ amd CTRL.
1. In equation (3), what is $Z^*$ here? And how is $\epsilon$ used in the right hand?
2. In equation (4), should $g(x, \eta)$ be $g(z, \eta)$?
3. In equation (7), should $min_{\theta}$ $max_{\eta}$ be $min_{\eta}$ $max_{\theta}$ ?

**Strength And Weaknesses:**

The good part of this paper lies in that the training process it proposed seems to be more stable than CTRL and achieves better performance.

Several concerns and questions I have are:
1. Limited novelty: Most of the designs (like the minimax game over coding reduction rate, the different roles of encoder, etc.) and equations of this paper are from CTRL, while this paper only slightly change the optimization design, i.e. optimizing both $\theta$ and $\phi$ in the minimizing part. So for me, instead of calling it a new generative model, I will only see it as an improvement that stablizes the training of CTRL.
2. Sensitive to hyperparameter: From Table 2, we can see that the performance of CoA-CTRL is highly sensitive to the CoA-ratio. Thus, although there may exist a better solution of CoA-CTRL than CTRL, carefully parameter tuning may be needed to reach this result, which may limited the usage of this technique.
3. How can new samples be generated after the model is trained? From figure 2 , it seems that the $z$ and $\hat{z}$ used in training are all inferred from observations $x$ and $\hat{x}$ instead of being directly sampled from a simple Gaussain distribution like GAN and no constraint for the prior distribution is applied to $z$ as the VAE. Then how can $z$ or $x$ be sampled after the model is trained? Also the paper only shows the reconstruction results. How are the generated samples look like?
4. The class-wise accuracy of CoA-CTRL in table 4 seems not be high enough. In fact, besides CTRL and catGAN , CoA-CTRL can not beat any other baselines. Consider the MNIST task is very simple, I would regard the performance differences as obvious and can not agree that CoA-CTRL achieves comparable results as its baseline models.
5. Besides on the simplest MNIST dataset, the sample reconstruction results shown in figure 4 and 6 are not good enough. Many figures are blurry and details are lost.
6. This paper proposes to cooperatively learn encoder and decoder, then how is this different/related to [1]. Also, as a model trained under new generative framework, besides GAN and VAE, other generative frameworks (like score-based model, diffusion model, energy-based model, etc.) should also be considered as baselines.

[1] Cooperative learning of descriptor and generator networks. PAMI, 2020

**Summary Of The Paper:**

This paper proposes an optimizatin technique (CoA-CTRL) to stablize the training of closed-loop transcription framework (CTRL). CTRL doing a minimax game of an encoder model $f(\theta)$ and a decoder model $g(\eta)$ over the coding rate reduction in the data feature space $Z$. The encoder in CTRL has two roles, the encoder role in Auto-Encoding and the discriminator role in GAN, while it is only optimized with the adversarial process.  CoA-CTRL thus proposed to optimize the encoder for both roles. This equals to first maximize $\theta$ over the coding rate reduction and then minimize $\theta, \eta$ over the same loss. The experiments suggest that this optmization design gives more stable training than CTRL and achieve better performance.

**Summary Of The Review:**

I think this paper proposes a technique that makes a more stable training under the CTRL framework. But some questions and concerns are there to be solved. Please check the Strength And Weaknesses for more details.

---

> ### Author Response · Authors · 2022-11-19
> **Response to Reviewer kFbE**
>
> We thank you for your valuable opinions. For your questions, we have addressed some in the general comments, and we want to address some further questions in this part.
>
> Q1
>
>
> A1:This is a good question. Although the change is small, its significance and necessity cannot be overlooked. We provide a new learning strategy for CTRL, i.e. cooperative adversarial learning, and this new learning strategy is necessary for CTRL to perform well.
>
> CTRL is a new framework which is an extension from Auto-Encoding and GAN. Original CTRL just follows the learning strategy of GAN, but does not take Auto-Encoding’s learning strategy into consideration. Although original CTRL contains segments of Auto-Encoding, its reconstruction is poor in terms of sample consistency. We believe that an adversarial learning strategy like GAN is not suitable for CTRL. Therefore, we introduce cooperative adversarial learning. Cooperative adversarial learning is unique and novel in that it utilizes encoder’s two roles in the learning strategy level, and it solves the original CTRL problems of sample-consistency, instability, and demand for big batchsize. Also, the fact that the change is small make cooperative adversarial learning easy to implement and integrated, and also resources-saving.
>
> In conclusion, Cooperative adversarial learning is a learning strategy consistent with CTRL and necessary for CTRL.
>
>
> Q2:
>
>
> A2:
>
> We believe the performance of CoA-CTRL (demonstrated by IS and FID) may be associated with Delta R, and CoA-ratio provides a method to adjust $\Delta R$ and keep $\Delta R$ in a stable curve. Therefore, we conduct experiments in Table 2 using different CoA-ratios to produces different $\Delta R$ to see the association between $\Delta R$ and CoA-CTRL’s performance. Besides, as Figure 6 in the revised paper shows, after adjustments by the CoA-ratio, different values of $\Delta R$ are all stable in the training process, and this is important for keeping learning in a stable way.
>
>
> Q3:
>
> A3:
> we follow generated methods of original CTRL,  $Z_{sample} = Z_{mean} + \alpha \sum_{i=1}^{r}n_{i}*\sigma_{i}*v_{i}$ where $ Z_{mean}$ is mean of features Z, $\sigma_{i}$ and $v_{i}$ is i_th singular value and principal components, $n_{i}$ is $\mathrm{iid}$ gaussian $\mathcal N(0,1)$, $\alpha$ is hyper-parameters
> You can see generated samples in Figure 8 in the revised paper
>
> Q4:
>
> A4:Good question. I think what is important is that cooperative adversarial learning improves original CTRL’s performance. The class-wise accuracy of the original CTRL is just about 89%, but CoA-CTRL is about 95%. The only difference between CoA-CTRL and the original CTRL is the learning strategy, but the improvement is quite clear.
>
> Q5:
>
> A5:
> There are some reasons for the blurry figures. Firstly. We just set the feature dimension (nz) at 128 to ease the experiments. Secondly, the pooling layer of Resnet is not good for encoder. We have adjusted the nz value and provides non-pool layer networks in the revised experiments. We also conduct additional experiments on ImageNet-1k to prove that our cooperative adversarial learning could improve sample-wise consistency in mechanism.
>
> Q6:
>
> A6:
> Thanks for the good suggestions. Cooperative adversarial learning focuses on the cooperation inside the system (i.e. between the encoder and decoder), but most of other cooperative methods focus on the cooperation between systems.
>
> mini errors:
>
> Thanks for your figure out errors, I have correct those
>
> $Z^*$  refer to transpose of $Z$
>
> $R(Z,\epsilon) = \frac{1}{2}\log \det(I+\alpha ZZ^*)$ where $\alpha = d/n\epsilon$

---

> > ### Comment · Reviewer_kFbE · 2022-11-27
> > **Still confused about the generated samples in figure 8**
> >
> > Dear authors:
> >
> > Thank you for your reply.
> > However, I still feel confused for the generation process of your model. The figures shown in figure 8 are grouped by each class and it seems that each subfigure group (a - f) are just variations from a certain figure (in the center column). Do you learn a seperate set of parameters for each class? If so, then this should be considered as conditional sampling with class label as supervision. But the baseline numbers in Table 2 seem to contain results from unconditional sampling, which makes it not a fair comparison. Or do you just "generate" samples by manipulate the latent space of a certain observed training image? If this is the case, then it would be more unfair for comparing it with the baselines. (Actually I'm not sure whether this can be considered as "generation".) Besides that, the choice of the number of principle components and $alpha$ seems to be very ad-hoc. You may need to retune this thing for each model trained on each dataset.

---

> > > ### Author Response · Authors · 2022-11-28
> > > **About generated processs**
> > >
> > > We appreciate your kind reply, here are some of our responses
> > >
> > > Q1:Do you learn a separate set of parameters for each class?
> > >
> > > A1:Answer is no. our experiments is conduct at a unsupervised setting, and in generated process we also conduct unconditional sampling for fair compare. Figure. 8 is just to shows the potential of unsupervised conditional generation, but not for comparison with other methods in table 2
> > >
> > > Q2:Do you just generate samples by manipulate the latent space of a certain observed train image?
> > >
> > > A2:Answer is no, we generated images consistent with original CTRL.
> > > we present Figure8 to show association of generative images and principal components rather than compare performance in this generated methods with other baseline in table 2. As Figure 8 show, generative images are high associated with principal component.
> > >
> > > Q3:Choice of principle components and alpha?
> > >
> > > A3:This question is very insightful. GAN and VAE generate from a defined distribution noise like N(0,1). But CTRL generate from distribution of Z, and distribution is unknown until we finish our training process. We generate images from distribution of features Z. Hypermeters alpha and number of principal are simple to set, more principal components always contribute to completeness of generated images, alpha could set at a consistent value in different data as it is a scale parameter

---

> > > > ### Comment · Reviewer_kFbE · 2022-12-06
> > > > **I want to keep my rating.**
> > > >
> > > > I want to say thank you to the author for their detailed responses. However, I still hold concerns about this work. As mentioned in my review, I think the novelty is limited since this work is just a improvement technique under the CTRL framework instead of a new generative model. Moreover, on one hand, although it outperforms CTRL, it fails to justify its advantage over other frameworks in task like class-wise accuracy or image reconstruction. On the other hand, although it shows good results on image generation task, I think its generation process is a little ad-hoc, since they first train the model and then manually pick the PCs and alphas. It might be the case that even for the same model, under different initializations or training process, these parameters may need to be adjusted. I don't think this paper is very bad that deserve a strong reject, but I also don't think its current version is good enough. Thus, I want to keep my rating as borderline reject.

---

### Author Response · Authors · 2022-11-19
**Clarification of motivation**

1.$\textbf{Our goal}$

Our goal is to explore a learning strategy that fits closed-loop transcription framework (CTRL) and could improve CTRL’s stability and sample-wise consistency.

2.$\textbf{Association between CTRL and cooperative adversarial learning}$

As our paper show, cooperative adversarial learning is extended from CTRL, but the focus is totally different. CTRL provides a closed-loop framework, and we provide a learning strategy for CTRL.
Different frameworks have different learning strategies. For example, Auto-Encoding applies a cooperative compress learning strategy via an encoder and a decoder; GAN follows the traditional minmax game in an adversarial process. CTRL provides a new framework which combines Auto-Encoding and GAN, but it just follows GAN’s minmax game learning strategy. It ignores Auto-encoding’s leaning strategy. Specifically, CTRL uses the encoder in two roles only in the level of framework structure, but in the level of learning strategy, it just uses the encoder as a discriminator.
We believe that minimax game may not be suitable for CTRL, which naturally unifies Auto-Encoding and GAN in terms of framework structure. Our cooperative adversarial learning unifies the learning strategies of Auto-Encoding (cooperative adversarial learning comprises a cooperative process) and GAN (cooperative adversarial learning comprises also an adversarial process). This learning strategy is consistent with features of CTRL.

3.$\textbf{Instability of adversarial learning and its incompatibility with deep networks (Why we use resnet18, resnet50 and resnet101 as our encoder?)}$
Adversarial learning provides an iteration learning signal, but it always faces instability problems, especially when using deep networks. Research in representation learning believes that deeper and bigger models have better performances. But for adversarial learning, deep and big models do not always generate better performances, or even collapse due to high power of big models. This is not ideal for further exploration and application of generative models. The instability problem builds a gap between deep networks and adversarial learning. Therefore, in the experiment, we employ Resnet18, Resnet50, and Resnet101 to show that even on these deep networks, learning process can be stable and active controllable, hoping to remove this gap.

---

### Author Response · Authors · 2022-11-19
**Advantages of cooperative adversarial learning compared with traditional regulations**

Cooperative adversarial learning is simple, controllable and suitable for closed-loop transcription. Regulation techniques always tend to limit the discriminator (sngan, wgan), but regulation techniques will always consume resources. Cooperative adversarial learning does not consume additional resources, and the implementation is easier and simpler than other regulation techniques.

Manageable control. Regulation techniques tend to limit discriminator, but how far should the limitation go difficult to decide and sometimes confusing. Cooperative adversarial learning provides a way manageable control by letting the discriminator or encoder provide a learning signal actively. In closed-loop transcription, the learning signal can be measured as $\Delta R$, which is distance between the real images and the transcribed images in features space. Based on cooperative adversarial learning, we can control CTRL’s $\Delta R$ at a stable level, therefore we can get a stable learning signal.

---

### Author Response · Authors · 2022-11-19
**Details of cooperative adversarial learning via closed-loop transcription**

We have revised the method part (3.2 COOPERATIVE ADVERSARIAL LEARNING: TWO ROLES, LEARN TWICE) in our revised paper to present more clearly.

---

### Decision · Program_Chairs · 2023-01-20

**Decision:**

Reject

**Justification For Why Not Higher Score:**

The current paper has issues about limited novelty and insufficient experiments. The AC could not accept the paper.

**Justification For Why Not Lower Score:**

N/A

**Metareview: Summary, Strengths And Weaknesses:**

The paper proposes a generative model with cooperative adversarial learning (CoA-CTRL). It is based on the existing closed-loop transcription (CTRL) framework. The CoA-CTRL has an encoder and a decoder. As to the adversarial learning, the encoder in CoA-CTRL serves as a critic to maximize the feature distance between the real data and the transcribed data, while as to the cooperative learning, the encoder and decoder cooperatively minimize the difference between the real data and transcribed data. The experiments suggest that the proposed CoA-CTRL can achieve better performance than CTRL. The advantage of the work is to presents an effective strategy to improve the CTRL framework, but the weakness is also obvious, that is, the contribution and novelty are incremental.  The paper receives a total of 5 review feedbacks (including 1 strong reject, 2 rejects and 2 marginally below the acceptance threshold). The AC downweights the review with a score of strong reject because it doesn't provide valid feedback. After the rebuttal, all the reviewers lean to reject the paper because of limited novelty (i.e., marginal contribution built on the existing CTRL framework) and unconvincing superiority compared with other generative models (i.e., lack of comparison with other state-of-the-art generative models). After an internal discussion, AC agrees with all reviewers that the current paper is not ready for publication, thus recommending rejecting the paper. AC urges the authors to improve their paper by taking into account all the suggestions provided by the reviewers, and then resubmit it to the next venue.